# Metabolomics and Network Pharmacology in the Exploration of the Multi-Targeted Therapeutic Approach of Traditional Medicinal Plants

**DOI:** 10.3390/plants11233243

**Published:** 2022-11-25

**Authors:** Bharti Sharma, Dinesh Kumar Yadav

**Affiliations:** 1Department of Pharmaceutical Sciences, College of Pharmacy and Health Sciences, St. John’s University, Queens, New York, NY 11439, USA; 2Department of Pharmacognosy, SGT College of Pharmacy, SGT University, Gurugram 122505, Haryana, India

**Keywords:** medicinal plants, metabolomics, phytochemical profiling, LC-MS, GC-MS, network biology, polypharmacology

## Abstract

Metabolomic is generally characterized as a comprehensive and the most copious analytical technique for the identification of targeted and untargeted metabolite diversity in a biological system. Recently, it has exponentially been used for phytochemical analysis and variability among plant metabolites, followed by chemometric analysis. Network pharmacology analysis is a computational technique used for the determination of multi-mechanistic and therapeutic evaluation of chemicals via interaction with the genomes involved in targeted or untargeted diseases. In considering the facts, the present review aims to explore the role of metabolomics and network pharmacology in the scientific validation of therapeutic claims as well as to evaluate the multi-targeted therapeutic approach of traditional Indian medicinal plants. The data was collected from different electronic scientific databases such as Google Scholar, Science Direct, ACS publication, PubMed, Springer, etc., using different keywords such as metabolomics, techniques used in metabolomics, chemometric analysis, a bioinformatic tool for drug discovery and development, network pharmacology, methodology and its role in biological evaluation of chemicals, etc. The screened articles were gathered and evaluated by different experts for their exclusion and inclusion in the final draft of the manuscript. The review findings suggest that metabolomics is one of the recent most precious and effective techniques for metabolite identification in the plant matrix. Various chemometric techniques are copiously used for metabolites discrimination analysis hence validating the unique characteristic of herbal medicines and their derived products concerning their authenticity. Network pharmacology remains the only option for the unique and effective analysis of hundreds of chemicals or metabolites via genomic interaction and thus validating the multi-mechanistic and therapeutic approach to explore the pharmacological aspects of herbal medicines for the management of the disease.

## 1. Introduction

Medicinal plants have been considered an essential source for drug discovery and development as they are enriched in verities of phytochemicals of different categories [1]. Due to the multiplicity of the phytochemicals, they have been associated with the multi-targeted therapeutic response and thus dealing with the alleviation of several acute and chronic ailments. However, over the past decades, the use of medicinal plants and their derivative products has seen rapid growth due to their minimal side effects, easy availability, and economically thus maintaining sustainable development in the health care system and economy of the country [2,3]. Furthermore, medicinal plants are accepted as the main source for new drug discovery and development, as more than approx. 50% of existing drugs or pharmaceuticals are derived from natural resources. A major contribution to their regulatory aspects includes taking a look at their uncountable use in the health care system, including quality, safety, efficacy-based evaluation, and the generation of scientific evidence. Based on pharmacological approaches, it is not easy to determine the exact principle of phytochemicals that are exhibited in the therapeutic response to the treatment of ailment, whether targeted or non-targeted [4]. It is widely distributed around the world in tropical and subtropical regions. Chemicals belong to the class of steroids, triterpenes, polyphenols, polysaccharides, proteins, etc. However, validating herbal medicines or products based on their quality, safety, and regulatory aspects is still a challenge.

However, several modern analytical techniques, including chromatography, nuclear magnetic resonance (NMR), and mass spectrometry (MS), are used abundantly in the quantitative and qualitative assessment of plant metabolites. However, these techniques are much preferred with plant phytoconstituents profiling or metabolomics studies due to their high sensitivity and accuracy when coupled with effective chromatographic techniques that allow separation and characterization of the diversity of phytoconstituents present in medicinal plants [5]. These techniques include gas chromatography-mass spectrometry (GC-MS), high-performance thin-layer chromatography (HPTLC), capillary electrophoresis–mass spectrometry (CE-MS), and liquid chromatography-mass spectrometry (LC-MS), and are most precise techniques used for quality control of medicinal plants [6].

In mass spectrometric analysis, *m*/*z* values of metabolites are matched to reference data from mass banks, where the analysis is resumed under a defined tolerance of *m*/*z* value variability. MS data acquisition relies on the probability score between the analysis query and the reference data search. For non-targeted metabolites, the data acquisition tolerance can be in both the *m*/*z* value and the chromatographic retention time, and further, these implications are accustomed to measurements using alignment software. Advanced web-based platform analysis was established based on automated workflows ranging from processing to the extreme alignment of row data, annotation, and statistical analysis tools to enable easy access to a wide range of investigators or informatics experts [7,8,9].

With the advent of the gradual rise of interdisciplinary disciplines related to bioinformatics and computational biology, researchers are becoming more motivated or are shifting from single-mode to multidisciplinary and dimensional systematic research modes for traditional medical research. There are important changes from the perspective of biomolecular networks involving the role of genes in disease regulation to understand the mechanism of drug action. The purpose of using “network computational biology” is to bring about significant change or to address new challenges associated with the initial testing of drugs. Network targeting represents unique concepts that characterize the biological network of diseases underlying therapeutic targets and understand the systematic mechanism of action for one of the multi-targeted drugs primarily for conventional medicine. Thus, the principle of “network prediction of targets” has been accepted as a fundamental principle of network pharmacology [10,11].

Traditional Medicine (TM), characterized by individualized, multi-component, and holistic medicine, holds great potential in conceptualizing the various challenges in the system of modern health care [12]. Traditional medicine (TM) has evolved from a single, isolated mode to a complex, methodical research methodology [13]. There is a lot of opportunity for traditional medicine (TM), which emphasizes holistic, individualized, and multi-component treatment, to help with some of the challenges in current healthcare. By using natural products as the primary molecule responsible for medication synergism and cumulative action, they aid in understanding novel therapies. These methods have been demonstrated to be effective in some herbal preparations used in conventional medicine [14,15].

Network pharmacology (NP), a brand-new area of study, aims to explain the interactions and actions of drugs with a variety of targets. It uses technology to systematically catalog a drug molecule’s molecular interactions with a living cell [16]. Research techniques for developing novel drugs are now using network-based methodologies. These methods have been demonstrated to be effective in some herbal products used in traditional therapies [17,18]. A current method for identifying active ingredients and possible molecular targets in a variety of herbal formulations or basic plants is known as network pharmacology [19,20].

Based on the phytopharmacological evolution of metabolites, the present study is associated with exploring the mechanistic molecular role of metabolites in the treatment of various diseases, including its properties such as antimutagenic, antiulcer, antioxidant, antidiabetic, antifertility, anthelmintic, antilipolytic, etc. In the molecular-based pharmacological assessment of metabolites, the present study explores the multi-targeted therapeutic approach of traditional Indian medicinal plants in the treatment of various acute and chronic diseases [21].

## 2. Methodology

Studies were conducted according to preferred reported items/checklists from systematic reviews or statements of research articles published in national and international journals.

### 2.1. Search Strategy

The information was collected through a comprehensive literature search using more than five electronic databases such as Science Direct, Elsevier, Google Scholar, ACS Publications, PubMed, and Springer from databases published between 2000 and 2022 or more recently for the study. Influential information was selected for targeted and untargeted data. Modern systems of medicine and therapy for treatment, traditional systems of medicine and medicinal plants for the treatment of disease, phytochemistry, nephroprotective activity, antioxidant activity, oxidative and anti-inflammatory activity, and inflammatory stress associated with the pathophysiology of diseases, etc., were all included in the review, and the research papers checked for potentially relevant citations in the reference list. The study selection was restricted to articles in English due to the language barrier, time efficiency, and high cost of translation. For comprehensive detection of relevant studies and recapitulation of the journal’s publication history, a systematic review was conducted with the hyphenated keyword “Role of medicinal plants in the management of diseases; A systematic review”. Appropriate data were extracted and summarized as prospective information on metabolites to reach the selection process.

### 2.2. Inclusion Criteria

Published reviews or research articles regarding the prevalence of diseases in national and international journals and metabolites represent a diverse group of low-molecular-weight structures, including lipids, amino acids, peptides, nucleic acids, and organic acids, that allow inclusive analysis to be combined into one.

### 2.3. Exclusion Criteria

Studies such as duplicate publications, abstracts, and research published before 2000, unauthentic ethnobotanical/ethnomedicinal reports lacking study areas/localities, informant’s involvement, data of untargeted plant and disease, and non-open access journal articles or in part accessed articles were removed to avoid miscomprehension.

### 2.4. Study Selection

All authors independently assessed the study through their inclusion criteria by examining the title and abstract of each record and obtaining its full text, if necessary. Study selection includes only well-indexed journal publications (research and review articles) related to diseases for trending input. The search results were screened based on the title and abstract of the identified journal articles/thesis. Furthermore, out of thousands of reports, only appropriate reports were downloaded and critically inspected to elaborate the discussion related to the preclinical and clinical evaluation of metabolomics and its exploration of multi-targeted therapeutic approaches of traditional Indian medicinal plants.

## 3. Review Findings

### 3.1. Need for Metabolomic Study for Medicinal Plants

We are aware of the multi-mechanistic and therapeutic action of medicinal plants and their derived products exhibited by the varieties of the metabolites present in the plant matrix, thus, alleviating several acute and chronic ailments. During this period, omics technology advancements were leveraged to create strategies for different aspects of drug research [22,23]. In order to explore the diversity of the metabolites in a plant matrix, there are various chromatographic and spectrometric techniques such as GC-MS, LC-MS, NMR, FT-IR, etc., which deal with the qualitative and quantitative exploration of the phytochemicals and each technique works based on their designed principles, hence exploring the complete phytochemistry phenomenon. The study of the medicinal effect of plants exponentially put many research orientations to the plant metabolites to obviate the insufficiency for medicinal plants or their derived products that evades them from meeting their worldwide requirements [24,25]. Plant secondary metabolites are one of the main principles for their quality, safety, and efficacy-based scientific validation; hence, they are exponentially used as the first choice of parameter for their regulatory aspects or authentication, although it has been reported that only safety and efficacy data is not sufficiently meeting the optimum criteria for their scientific evaluation and health prospects [26]. Throughout history, traditional medicines have been used as folk regimens, and in recent past decades, these are providing a key role in the production of many natural products to treat a variety of ailments. It has been reported that secondary plant metabolites play an essential role as exogenous and endogenous agents for treating varieties of pathogens and thus producing a large-scale antibiotic only by the plant sources, which significantly act against the different microorganisms which cause several pathophysiological changes or health issues [27,28]. Based on the multi-mechanistic and therapeutic effect of the plant secondary metabolites, these have been termed as the economical as well as the potential therapeutic weapons for treating several ailments induced by the exogenous and endogenous PAMPs or DAMPs onsets.

#### Challenges in Metabolomics Analysis

It is known that medicinal plants deal with the diversity of metabolites even with different chemical or structural behavior. Furthermore, to reach perfection in exploring the phytochemistry of medicinal plants, a single analytical method does not put into their high throughput phytochemical screening [29]. As each technique works based on its designed principle, multiple methods or techniques are required to explore qualitative and quantitative phytochemical profiling in a medicinal plant [30]. Some approaches need to be thought of during metabolomics studies using modern analytical techniques, such as the extraction of phytochemical constituents from a plant matrix. It is processed based on their ethnopharmacological prospective or targeted bioassay, so it is necessary to extract phytochemicals using traditional or modern reference, understand the phytochemical behavior of the targeted or untargeted metabolites such as probable solubility/polarity, chemical nature of molecule (acidic or basic) and complexity of the molecule/molecular size. Furthermore, in metabolomic analysis, the preparation of the sample is one of the essential steps which support the whole outcome of the study. The targeted solvent and its chemical behavior, solvent ionization followed by its dielectric behavior, and sample amount need to be appropriate for accurate and precise analysis. Chromatographic conditions, mobile and stationary phase (nature of stationary phase) used for separation of the targeted or untargeted metabolites should be followed based on the chemical nature of the metabolites. Last but not least, the ionization process, source of ionization, and flighting distance and time need to be adjusted and favored also based on the chemical nature and complexity of the metabolites. Moreover, the repeatability of the optimized or non-optimized chromatographic and spectroscopic methods needs to be validated to reproduce robustness [31].

Based on the above facts and assessment for metabolomic analysis helps us to unravel and even reach perfection in qualitative and quantitative phytochemical analysis of medicinal plants. Furthermore, in a single study, quantitative-based standardization of the medicinal plant needs at least four marker constituents to validate the authenticity aspects for their regulatory purpose [32]. A systematic method for metabolomic approach and statistical analysis has been depicted in Figure 1 and previously reported methods for quality and quantitative metabolomic analysis of medicinal plants using the modern analytical technique are summarized in Table 1.

### 3.2. Needs of Network Pharmacology for Therapeutic Exploration of Medicinal Plants

Computational techniques have been growing and exotically contributing to drug discovery and development via assessing wide varieties of genes and their expression behavior to the chemical or bioactive principles which are involved in the pathophysiology of any ailments. Compounds and protein interaction and biomolecular accession provide a high throughput orientation to explore better pharmacological validation concerning the targeted or untargeted metabolites. Besides this approach, the one-drug/one-target/one-disease strategy for drug development is currently confronted with several difficulties in terms of sustainability, efficacy, and safety. Recently, strategies for network biology and pharmacology, respectively, have acquired recognition [33,34].

Network biology and polypharmacology are one of the most conventional approaches to exploring multi-mechanistic and therapeutic targets for integrating omics data and developing multitarget drugs. Hence, both approaches are combined and acknowledged as network pharmacology. This approach does not look only at the consequence of drugs in terms of the effect on both the interactome as well as the diseasome level and covers multiple interactions with multiple proteins/genes in a single time [18]. By enabling an objective examination of prospective target areas, it also seeks to identify novel drug leads and targets and to repurpose current drug molecules for various therapeutic diseases [35]. Traditional knowledge about the plant and its omics plays a vital role in the further discovery, development as well as repurposing of existing drugs. However, this approach requires the right protocol and guidance for screening the right targets and new scaffolds which are being used for the evaluation of drug molecules. During the analysis of the network biology or pharmacology, the rationale design and therapeutic targets should be known (Cho et al., 2012, Hopkins, 2008, Ellingson et al., 2014) [33,34]. Such open new therapeutic options have also aimed to improve the safety and efficacy of existing medications. Network pharmacology analysis has been generally classified into two different approaches such as untargeted network pharmacology analysis and target-based network pharmacology analysis. A systematic approach for network biology as well as polypharmacology has been depicted in Figure 2.

#### 3.2.1. Untargeted Network Pharmacology Analysis

In this analysis, the active molecule can be fixed or can not be fixed, which needs to be explored biologically based on the interaction of targets/genes targets that would be interacted with the molecule are completely unfixed. A random analysis is done using the software commands (Cytoscape) of the STRING app to expand the network. During this process, the number of interactors to expand the network depends on the information retrieval that exists in the database (https://jensenlab.org/, 19 October 2022), while the type of interactors to the expanded network should remain to the *Homo sapiens* species only. The process is repeated till no interactors are further connected via expanding the network. No further expansion of the network represents no additional genomic information exists in the reported database. Thereafter, the statistical information of the generated network about the number of nods, edges, network density, heterogeneity, etc., is imported through a network analyzer. In one example, the constructed network with statistical summary has been depicted in Figure 3.

#### 3.2.2. Target Based Network Pharmacology Analysis

In this analysis, the active components and each target are fixed already concerning the disease. During this analysis, a set of active components and genes involved in the pathophysiology of the disease are simultaneously analyzed for their interaction in the form of nods and edges. The genes are generally screened for the Genecard database (genecards.org), which is a free online database that involves an n-number of genes involved in diseases. STITCH is an open-source platform for rapid analysis of the active components and the genes which are needed to be analyzed. Each component and selected gene are inserted into the STITCH platform of the Cytoscape software. It gives multiple possible matches at the fixed or optimum confidence score of 0.04 and maximum additional interactor [34,36]. During network analysis, several additional interactors were found to be connected with the genes, which are generally removed through network interpretation analysis. Statistical summary, betweenness among the network, and strength of the network binging between the nodes as gene and the active component are evaluated as network data acquisition or summary of the network. Furthermore, there are various new approaches, such as a network histogram analysis which gives information about the enrichment of the genes biologically active or based on cellular organelles or tissue. Multi-statistical components analysis, principal components analysis, principal coordinates analysis, t-distribution analysis, etc., is performed to retrieve information about the biological enrichment of the gene or based on their pathophysiological actions. An example of the analysis has been subjected in Figure 3.

Previously, several studies have been reported based on the targeted and untargeted network pharmacology analysis, which revealed the multi-mechanistic and therapeutic role of various biologically active metabolites in the alleviation and treatment of diseases. In a study reported by Gautam evaluated multi-mechanistic and therapeutic potential of *Momordica charantia* in the alleviation of diabetic nephropathy. In this study, the author explored the therapeutic potential of several metabolites of *Momordica charantia* and reported that quercetin, ferulic acid, caffeic acid, and catechin are the potential metabolites that exhibit a significant interaction with the genes such as G6PD, MAPK1, MAPK3, NOS, PTGS2, PON1, ILs, etc. and alleviating diabetic nephropathy and its associated complications [34]. Gaurav et al. reported the multi-mechanistic and therapeutic potential of polyphenols present in *Boerhaavia diffusa* and *Tinospora cordifolia* against glomerulopathy, inflammation, oxidative stress, etc. The outcome of the study showed that *Boerhaavia diffusa* and *Tinospora cordifolia* contains several metabolites, such as polyphenols which poly a significant role in the alleviation of kidney disease and its associated complication via the regulation of several genomes such as CASP3, CASP7, CASP8, CASP9, MAPK1, MAPK3, TNF and TP53 involved in the several pathways AGE/RAGE pathway, signaling to interleukins, response to the inorganic substances, TNF signaling pathway, lipid localization, regulation of defense response, RAS and bradykinin pathway in COVID-19, circulatory system process, cellular response to nitrogen compounds, etc. [33].

#### 3.2.3. Gene Ontology or Gene-Disease Association Network Analysis

Gene ontology analysis showed the multi-targeted and mechanistic approach of the targeted genes that are screened through the protein-protein network (PPI) and compound-protein interaction (CPI) network analysis. it gives information about that how genes play multi-targeted effect in pathophysiology involved in the disease via regulation of several positive and negative biological pathways. There are several platforms such as Cytoscape, network analyst (https://www.networkanalyst.ca/, 19 October 2022), Metascape (https://metascape.org, 19 October 2022), STRING (https://string-db.org/, 19 October 2022), DisGeNET (https://www.disgenet.org/, 19 October 2022), etc. outlined in Table 2. These are the freely available gene analysis tools for gene ontology analysis and give enriched information about gene-gene interaction and their involvement in diseases. The genes which are involved significantly in the alleviation of the disease through any specific pathway are also revealed through gene ontology analysis which is expressed in form of the p-value. Furthermore, the gene enrichment analysis also reproduces a robust approach for biological or physiological regulation of the plant growth or production of secondary metabolites [33,34]. A systematic approach for the evaluation of gene and disease association has been depicted in Figure 4.

Gupta et al. evaluated functional annotation and comparative analysis of *Withania somnifera* Leaf and root transcriptomes to identify putative genes involved in the withanolides biosynthesis through various approaches of gene enrichment analysis. The outcome of the study showed that, to assign functional GO annotation in terms of biological process, molecular function, and cellular component groups, the TAIR10 database’s best hit for each unigene. The wide range of GO annotations for unigenes emphasizes the variety of genes that are probably represented in the transcriptome data for the Withania leaf and root. A significant number of unigenes are involved in the production of several secondary metabolites by mapping these unigenes onto KEGG. It was also found that the genes for all of the enzymes involved in the biosynthesis of the triterpenoid backbone (including the MVA and MEP routes) up to 24-methylene cholesterol, which is thought to be a precursor for the biosynthesis of withanolide, based on the annotation. Additionally, certain contigs associated with the enzymes used in these processes displayed differential expression [37].

Exploration of multi-mechanistic and therapeutic action of traditional medicines through network biology, polypharmacology and gene ontology analysis has been increased exponentially. Vadivu et al. reported that mapping genes involved in the diseases with standardized ontology analysis gives more improvement in analyzing the biological performance of the medicinal plants and their therapeutic options [38]. Several studies have been conducted to investigate multi-targeted effects or molecular mechanisms of some herbal medicines or their derived products using Network pharmacology approaches (Table 3). Gaurav et al. reported the multi-therapeutic use of two Indian traditional medicinal plants such as *Boerhaavia diffusa* and *Tinospora cordifolia,* in the regulation of several pathways such as signaling to interleukins, response to the inorganic substances, TNF signaling pathway, lipid localization, regulation of defense response, RAS and bradykinin pathway in COVID-19, circulatory system process, cellular response to nitrogen compounds, etc. [33]. Gautam et al. validated the traditional pharmacological claim of *Momordica charantia* in alleviating diabetic nephropathy and its associated complications via regulation of several pathophysiological pathways such as metabolic regulation, increased uptake of glucose to the muscle, obviating insulin resistance, biochemical markers regulation, oxidative stress, and inflammation, etc. [34].

*Physostigma venenosum* potential targets were identified using a combination of network pharmacology and virtual reverse pharmacology techniques for the treatment of vascular dementia (VaD). This work used PASS prediction to find known VaD-related targets interacting with phytomolecules. More than 200 proteins connected to VaD were studied since its expression has increased. In order to broaden the scope of the search, an extra 34 genes were chosen. It was done because, from a pharmacological perspective, it is typically simpler to inhibit the function of proteins than to activate them. Thirty-four mechanisms of action (MOAs) predicted by the PASS program were found to exist. It was found that eseridine, physostigmine, physovenine, and eseramine can be predicted with a probability Pa of more than 0.5 for their dual MOAs (inhibition of acetylcholinesterase and butyrylcholinesterase) and pharmacological effects related to the treatment of VaD (cognition disorder treatment and nootropic). Acetylcholinesterase and butyrylcholinesterase are both inhibited by eseridine42. Physostigmine holds patents for treating dementia and Alzheimer’s disease as well as inhibiting butyrylcholinesterase, acetylcholinesterase inhibition, and memory enhancement. While eseramine has a less inhibitory effect against acetylcholinesterase than physostigmine, physovenine was patented as an inhibitor of acetylcholinesterase and butyrylcholinesterase as well as for the treatment of AD and dementia. The anti-acetylcholinesterase and anti-butyrylcholinesterase activity of several phytomolecules may have additive or synergistic effects on the therapeutic benefits of Physostigma venenosum in the treatment of VaD. It is now understood that this plant is utilized to cure atherosclerosis [39].

Khanal et al. performed gene ontology enrichment analysis of α-amylase inhibitors from *Duranta repens* (*D. repens*) in diabetes mellitus. Through network pharmacology analysis between -amylase inhibitors, regulated proteins, and expression pathways, it was possible to determine the -amylase inhibitory action of substances from Duranta repens. In the compound-protein-pathway network, it was discovered that the PI3K-Akt signaling route and the p53 signaling pathway were anticipated to be substantially regulated. Scutellarein was also projected as the main hit based on its ability to inhibit -amylase, its affinity for binding, and its involvement in regulated pathways. Additionally, AGE-RAGE and the FoxO signaling pathways, which are implicated in diabetic complications, were projected to be modulated by -amylase inhibitors. It was shown that *D. repens* may not only block -amylase in the GI tract, but may also be absorbed into the systemic circulation and influence several pathways implicated in the etiology of diabetes mellitus to generate a synergistic or cumulative impact [40].

Traditional Indian medicinal plants and their therapeutic exploration through network pharmacology. Research techniques for developing novel drugs are increasingly using network-based methodologies. By using natural products as the primary molecule responsible for medication synergism and cumulative action, they aid in understanding novel therapies. These methods have been demonstrated to be effective in several herbal preparations used in conventional medicine [17,41,42,43].
plants-11-03243-t001_Table 1Table 1Quality and quantitative metabolomic analysis of medicinal plants using the modern analytical technique.Sr. NoMedicinal Plant and Part Used for AnalysisTechniqueMethod/Type of AnalysisIdentified Metabolites (Major)References1*Achyranthes aspera* (Leaf)LITE-5MS GC-MS (Qualitative)Column: 30 m × 0.25 mm I.D., 0.25 µmMedium: Helium as carrier gas, flow rate of 1.1 mL/minButanoic acid, Phytol, 1-Butanol, 6-Octen-1-ol, 3,7 Dimethyl-, Propanoate, Decanoic acid, 6-Methylfuro [2,3-C]pyrid-5-one, 3,7,11,15-Tetramethyl-2-hexadecen-1-ol, 2-Decen-1-ol, 3-Bromo-2-Methoxycyclohexanone, 2-Nonen-1-ol, Cyclohexane, 1-Hexadecyne (14.43%), 6-Octen-1-ol, Propanoate, 6-Octen-1-ol, 3,7 Dimethyl-,Propanoate.[44]2*Aerva javanica* (Leaf)GC-MS (Qualitative)Column: HP-5MS(30 m × 0.025 mm)Medium: Helium as a carrier gas, flow rate of 13 mL/min3-allyl-6-methoxyphenol, dodecanal, (E)-6,10-dimethyl-5,9-undecadien-2-one, trans-β-ionone, β-panasinsene[45,46]3*Aerva lanata* (Aerial parts and roots)LC-ESI-MS/MS(Qualitative and quantitative)Column: (30 m × 0.25 mm I.D., × 1 EMdf) Medium: Helium as carrier gas, flow rate of 1 mL/min.Quercetin, kaempferol, and myricetin, Gallic acid, *p*-coumaric, Vanillic, Caffeic acid, Rutin, Ferulic acid, Astragalin[47]4*Allamanda cathartica* (Leaf)GC-MS (Qualitative)Column: 30 mm in length, 0.25 mm I.D., and 1 µmMedium: Helium as a carrier gas, flow rate of 1 mL/minGlycerin, n-hexadecanoic acid, Phytol, Thymine, Tetradecanoic acid, Dodecanoic acid, Octanoic acid, ethyl ester, [48,49]5*Artemisia absinthium*(Aerial part)GC/MS HS-SPME analysis(Qualitative and quantitative)Colum: DB-5(30 × 0.2 mm, film thickness 0.32 µm)Medium: helium gas, flow rate of 1.7 mL/minCamphor, p-cymene, Isoledene, Caryophyllene, Isopulegol Acetate, Hysterol, Isocaryophillene, Diisoamylene, β-farnesene, and Cyclohexane,2,4-diisopropyl-1,1-dimethyl[50]6*Aurantii fructus* (Dry whole fruit part)HP-5 MS (Qualitative and quantitative)Column: (30.0 m × 250 μm × 0.25 μmMedium: Helium as a carrier gas, flow rate of 3 mL/min*p*-Xylene, (−)-*α*-Pinene, α-Phellandrene, 3-Carene, d-Limonene, Ocimene, 4-Carene, Linalool, Terpineol, Limonene oxide[51]7*Bergenia ligulata* (Rhizome)GC-MS (Qualitative)Colum: 100 m × 0.25 mm. I.D., 0.5 μm, Film thickness: 0.25 μm.Medium: Helium as a carrier gas, flow rate of 1 µL/minTadecanoic acid methyl ester, hexadecanoic acid, methyl ester, quinoline and phenol-2,4-bis (1,1- dimethylethyl), hexadecanoic acid, methyl ester, octadecanoic acid methyl ester, and 2-Propyl-5-oxohexanoic acid, aR Turmerone, 3,7,11,15-Tetramethyl-2-hexadecen-1-ol and Squalene[52]8*Boerhaavia diffusa* (Root and whole plant)GC-MS (Qualitative and quantitative)Colum: 30 m × 0.25 mm I.D., × 0.25 μm Medium: Helium as a carrier gas, flow rate of 1 mL/min Sucrose, l-Tyrosine, Malic acid, Uracil, Succinic acid, Fumaric acid, 4-Methylcatechol, D-Pinitol, [53,54]9*Boerhavia diffusai* (Root and aerial parts)
Column diameter is 0.32 mm; column length is 30 m, column thickness 0.50 μm), Medium: Helium gas, flow rate of 1.73 mL/minBoeravinone-B, eupalitin galactoside[55] 10*Carica papaya* (Leaf)GC-MS QP-2010 ULTRAColumn: 30.0 m × 0.25 mm × 0.25, Medium: Helium as carrier gas, flow rate of 1 mL/min.Myricetin, caffeic acid, trans-ferulic acid, and kaempferol[56]11*Chlorophytum borivilianum* (Tuber)GC-MS (Qualitative and quantitative)Column: 25 m × 0.25 mm I.D., × 0.25 umMedium: Helium as carrier gas, using 122.2 KPa (51.6 cm/s)Chlorophytoside-I (3β, 5α, 22R, 25R)-26-(β-D-glucopyranosyloxy)-22-hydroxy-furostan-12-one-3 yl O-β-D-galactopyranosyl (1-4) glucopyranoside[57]12*Chrysanthemum morifolium* (Flower)LC-DAD-ESI/MS (Qualitative)Column: 30 m × 0.32 mm × 0.5 µmMedium: Helium as carrier gas, flow rate of 1.5 mL/minβ-Humulene, Ledene oxide-(I), Caryophyllene, Eicosane, Heneicosane, Germacrene, Limonene, Borneol, α-Farnesol, Camphene etc.[58]13*Cichorium intybus* (Root)Rtx-5 MS GC-MS (Qualitative)Column: 30.0 m × 0.20 mm I.D., 0.25 μmMedium: Helium as carrier gas, flow rate of 1 mL/min2-methoxy phenol, 2,3-butanedione, 2-furfurylthiol, 2-thenylthio, 1-octene-3-one, 2-ethyl-3,5-dimethylpyrazine, 3-methylbutanal, 2,3- butanedione[59,60]14*Dicentra spectabilis,*HP-5 MS GC-MS (Qualitative and quantitative)Column: 30 m × 0.25 mm × 0.25 μmMedium: Helium as a carrier gas, flow rate of 1 mL/min Hordenine, Trisphaeridine, Hamayne, Caranine, Galanthine, Homolycorine, Galwesine[61]15*Didymocarpus pedicellata* (Leaf)GC/FID and GC/MS (Qualitative)Column: 30 mm in length, 0.25 mm i.d., and 1 µmMedium: Helium as carrier gas, flow rate of 1 µL/minβ-caryophyllene, α-humulene, β-selinene and α-selinene, caryophyllene oxide, veridiflorol, spathulenol, and humulene oxide[62]16*Dracocephalum moldavica*, *Ocimum americanum*, *Lophanthus anisatus*, *Monarda fistulosa*, and *Satureja hortensis* (aerial parts)HP-5MS GC/MS Colum: HP-5 column (30 m × 0.25 mm i.d with 0.25 μm film thickness)Medium: Helium as carrier gas, flow rate of 1 mL/min.Geraniol, geranyl acetate, thymol, citral, β-caryophyllene[63]17*Fragaria vesca* (Berries)GC-MS (Qualitative and quantitative)Column: 0.25 mm ID 15 m length 1.0 µmMedium: Helium as carrier gas, flow rate of 1 mL/minEugenol, δ-Decalactone, Hexanoic acid, Terpinen-4-Ol, ϒ-Hexalactone, Ethyl pentanoate, Phenylethyl acetate[64]18*Genista tinctoria*(NA)HILIC (Qualitative)
Genistein, luteolin, naringenin, quercetinmyricetin, apigenin, quercitrin[65]19*Glycyrrhiza inflate* and *Glycyrrhiza echinata* (Root)GC-MS (Qualitative)Column: (30 m × 0.32 mm × 0.25 μm)Medium: Helium as carrier gas, flow rate of 1 mL/minCadaverine and myo-inositol[66]20*Glycyrrhizae radix* (Root)GC-MS (Qualitative and quantitative)Column: 30 m 0.25 mmMedium: Helium as carrier gas, flow rate of 1 mL/minα-bisabolol, eudesmol, β-Pinene, Myrcenol, Germacreme, Limoneon monoxide, D-Limonene, β-Citronellal, Tricyclene[67]21*Gymnema sylvestre* (Leaf)GC-MS analysis was conducted using SHIMADZU, QP2010 PLUS(Qualitative)Column: 29.3 m × 0.7 m, 320 μm. Medium: Helium as carrier gas, flow rate of 1.5 mL/minSqualene, Tetratriacontane, Phytol, n-Hexadecanoic acid, Phthalic acid, di(2-propylpentyl) ester, Benzoic acid, 3,5-dicyclohexyl-4-hydroxy-, methyl ester[68]22Hydrophilic andhydrophobic compounds mixtureHILIC-RPLC (Qualitative)
Pseudouridine, isonicotinic acid, palmatine, uracil, adenosine, nicotine, propranolol,Atenolol, 4-hydroxybenzoicAcid, hippuric acid, phenol, anisic acid, 4-nitrobenzoic acid, 4-chlorobenzoic acid, 2,6-dimethyl phenol, 2,3-dimethyl naphthalene, 4-chloro biphenyl, fluoranthene, pyrene.[69,70]23*Lepidium sativum*TR5–MS GC-MS (Qualitative)Column: (30 mm × 0.25 mm I.D., × 1 μMdf)Medium: Helium as carrier gas, flow rate of 1 mL/minEugenol, tigmastane-3,6-dione, n-Hexadecanoic acid, Dodecanoic acid, n-Hexadecanoic acid, Stigmasterol, etc.[71] 24*Lolium perenne* (Stem)HP-5 MS GC–MS (Qualitative and quantitative)Column: 30 m × 0.25 mm × 0.25 µmMedium: Helium as carrier gas, flow rate of 1 mL/minCamphene, p-cymene, bornyl acetate, amorph-4-en-7-ol, cadinol, bisabolol, amorpha-4,7(11)-dien-8-one and 3-acetomorpha-4,7(11)-dien-8-one[72]25*Lupinus angustifolius* (Aerial parts)HP-5 GC-MS (Qualitative and quantitativeColumn: 30 m 0.25 mm I.D., 0.25 μm)Medium: Helium as carrier gas, flow rate of 1 mL/min13-tigloyloxylupanine, lupanine 11,12-dehydrosparteine, and tetrahydrorhombifoline, 13-tigloyloxylupanine, tetrahydrorhombifoline, and lupanine[73]26*Mangifera indica* (Leaf)GC-MS (Qualitative and quantitative)Column: PB-1 column (Supelco, USA, 30 × 0.32 mm I.D., DF = 1 μm)Medium: Helium as carrier gas, flow rate of 1 mL/minα-thujene, camphene, R-terpinene, limonene, linalool, R-terpineol, eugenol[74]27*Mentha pulegium* and *Origanum majorana* (Aerial parts)GC-MS (Qualitative and quantitative)Column: HP5LS (30 m × 0.025 mm)Medium: Helium as carrier gas, flow rate of 2 mL/minα-pinene, β-pinene, 3-Octanol, Methyl cyclohexene, Menthone, Iso-menthol, Piperitenone, Dodecane, 2, 2-dimethyl propylidene[75]28*Morinda officinalis* (Root)HP-5 GC-MS (Qualitative and quantitative)Colum: 30 m × 0.25 mm I.D., with 0.25 μmMedium: Helium as carrier gas, flow rate of 1 mL/minTecnazene, alpha-BHC, Hexachlorobenzene, beta-BHC, quintozene, gamma-BHC, delta-BHC, heptachlor, octachlorodipropyl ether, chlorothalonil, triadimefon, aldrin, dicofol, fenson, fipronil, chlorfenvinphos, heptachlor epoxide,[76]29*Moringa oleifera* (Leaf)GCMS (Qualitative and quantitative)Colum: HP-5MS (30 m × 0.25 mm I.D., × 0.25 µm)Medium: Helium as carrier gas, flow rate of 1 mL/minPropanamide, D-Mannoheptulose, N-Isopropyl-3-phenylpropanamid, 1,3-Propanediol, 2-ethyl-2- (hydroxymethyl), Propionic acid, 2-methyl-, octyl ester, Ethanamine, N-ethyl-N-nitroso, and 9,12,15-Octadecatrienoic acid,[77]30*Morus alba* (Leaf)HP-S GC-MS (Qualitative and quantitative)Column: 30 m × 0.25 mm × 0.25 µmMedium: Helium as carrier gas, flow rate of 1 mL/min9,12,15-octadecatrienoic acidethyl ester, linolenic acid ethyl ester, gibberellic acid 4-methoxy phenol, ethyl isoallocholate, and octadecanoic acid[78]31*Musa paradisiaca* (Fruit pulp oil)HP-5 MS GCMS (Quantitative)Column: (30 m × 0.25 mm × 0.25 μm. Medium: Helium as carrier gas, flow rate of 1 mL/min.α-Thujene, γ-Terpinene, α-Pinene, 2-β-Pinene, Limonene, Butanoic acid, α-Terpinene[79]32*Myristica fragrans* (Whole parts)GC-MS (Qualitative and quantitative)Colum: 30 m × 0.25 mm I.D., × 0.25 µL Medium: Helium as carrier gas, flow rate of 1.23 mL/minElemicin, isoelemicin, myristicin, surinamensin, malabaricone C, 2-(3′-allyl-2′,6′-dimethoxy-phenyloxy)-1-acetoxy-(3,4-dimethoxyphenyl)-propyl ester, methoxylicarin A, licarin A, malabaricone B, licarin C, 5′-methoxylicarin B, licarin B, and 2-(3′-allyl-2′,6′-dimethoxy-phenyloxy)-1-methyl-5-methoxy-1,2-dihydrobenzofuran[80]33*Nardostachys jatamansi* (Root)GC-MS-HT-TOF(Qualitative and quantitative)Column: 29.3 m × 0.7 m, 320 μm. Medium: Helium as carrier gas, a flow rate of 1.5 mL/minDodecane, Linalool, l-calamenene, A-ionone, Oleic acid, Palmitic acid, Heptacosane, a-Muurolene, Tridecanoic acid, methyl ester.[81]34*Ornithogalum procerum* (Aerial parts)GC-MS (Qualitative)Column: (60 m × 0.25 mm × 0.25 μm)Medium: Helium as carrier gas, flow rate of 1 mL/minLinalool, Nonanal, γ-Terpinene, Octanal, Hexanal, Camphene, Furfural[82]35*Paederia foetida* (Leaf)GC-MS and NMR (Qualitative and quantitative)Column: (30 m × 250 μm × 0.25 μm).Medium: Helium as carrier gas, a flow rate of 1 mL/min.1,3,5-benzenetriol, palmitic acid, cholesta-7,9(11)-diene-3-ol, 1-monopalmitin, β-tocopherol, α-tocopherol, 24-epicampesterol, stigmast-5-ene, 4-hydroxyphenylpyruvic acid, and glutamine[83]36*Potentilla anserine* (Whole plant) HP-5MS GC-MS (QualitativeColumn: 30 m × 0.25 mm × 0.25 µmMedium: Helium as carrier gas, flow rate of 1 mL/minGlycolic acid, Alanine, Valine, Threonine, Malic acid, Aspartic acid, Shikimic acid, Pinitol, Quinic acid, etc.[84]37*Psidium guajava* (Leaf)GC-MS (Qualitative and quantitative)Column: 30 m × 320 µm × 0.25 µmMedium: Helium as carrier gas, flow rate of 3.3245 mL/minascorbic acid, α-tocopherol, 3-hydroxyanthranilic acid, Cis-Zeatin-9-glucoside, Quercetin, 5-O-Galloylquinic acid, Ellagic acid Kaempferol, Methylsuccinic acid, Oxysterol (R)-3-Amino-4-phenylbutyric acid, etc.[85]38*Rostellularia diffusa* (Whole plant)GC-MS (Qualitative)Column: (30 m × 0.32 mm × 0.25 μm)Medium: Helium as carrier gas, flow rate of 1 mL/min16-Hentriacontanone (22.59%), Hexadecanoic acid (11.23%), Stigmast-5-en-3-ol (6.78%), 9-Octadecenoic acid (n = 40)[86]39*Salix alba*GC-MS (Qualitative)Column: 60 M TRX 5-MS (30 m × 0.25 mm × 0.25 µm), Medium: Helium as carrier gas, flow rate of 1.21 mL/minTerpineol, 3-Thujanol, Eicosanoic acid, methyl ester, 1-Octadecyne, 1,2,3-Propanetriol, 2,2-Dimethylbutane, Acetic acid, butyl ester[87]40*Scutellaria barbata*
2-D HILIC × HILIC-MS (Qualitative)Naringin, luteolin-7-O-glucoside, apigenin-7-O-glucoside, 5,7,8,2′-tetrahydroxyflavone-7-O-glucoside, scutellarin, carthamidin-7-O-glucuronide/isocarthamidin7-O-glucuronide, luteolin-7-O-glucuronide,[88,89]41*Solanum nigrum* (Aerial parts)GC-MS (Qualitative and quantitative)Colum: HP-5 column (30 m × 0.25 mm with 0.25 μm film thickness)Medium: Helium as carrier gas, flow rate of 1 mL/min.4-nitroguaiacol, 3-cyclohexen-1-ol,4-methyl-1-(methylethyl), Nonanoic acid,1-methyl ethyl ester, Flavone, 5Decenedioic acid,5,6dimethyl, dimethyl ester, Oleic acid, Z, E-2-methyl-3,13-octadecadien-1-ol, 2,4-ditertbutylphenol, 2,6,6,10-tetramethylundeca-8,10-diene-3,7-dione[90]GC-FID-MS (Qualitative)Column: 30 m × 0.25 mm; 0.25 μm Medium: Helium as carrier gas, flow rate of 1 mL/minThymol, Carvacrol, Eugenol, Tetradecanal, iso-Longifolol[91]42*Sugarcane molasses*(Plants)Rtx-5MS GC-MS (Qualitative)Column: 30 m × 0.25 mm × 0.25 µmMedium: Helium as carrier gas, flow rate of 1 mL/minEthylene glycol, Lactic acid, Acetic acid, Glyceric acid, Glyceric acid isomer, Ribitol, Glucitol, Malic acid, Citric acid, Threonic acid, Fructose, Inositol, etc.[92]43*Swertia chirata* (Leaf)Elite-5MS LC-MS (Qualitative and quantitative)Column: 30 m × 0.25 mm I.D., 0.25 µmMedium: Helium as carrier gas, flow rate of 1.1 mL/minXanthone, Succinic acid, Viminalol, O, N-Permythylated N-Acetyllsine, Hexadecanoic acid, Cis-13-Octadecenoic acid, Tetracosanoic acid[93]44*Tagetes erecta* (Flower)GC-MS (Qualitative and quantitative)Colum: HP-5MS (30 m × 0.25 mm ID × 0.25 µm)Medium: Helium as carrier gas, flow rate of 0.56 mL/minCaryophyllene, Germacrene, Spathulenol, Palmitic acid, Heptadecanoic acid, Linolenic acid-metil ester, Bicyclogermacrene[94]45*Tarconanthus camphorantus* (Leaf)DB-1 GC-MS (Qualitative and quantitative)Column: (0.25 µm film × 0.25 mm I.d. × 30 mMedium: Helium as carrier gas, flow rate of 1 mL/mincarvacrol, tetracontane, squalene, tetrapentacontane, and Phytol[95]46*Theobroma cacao*(Seed)GC-MS (Qualitative)Column: 30 m × 0.25 mm × 0.25 µmMedium: Helium as carrier gas, flow rate of 1.5 mL/minDimethyl sulfone, 2-Cyclohexane-1-one, (1-methyl), hexadecanoic acid, methyl ester, Hexadecanoic acid, Octadecanoic acid,[96]47*Tinospora cordifolia* (Root)GC-MS (Qualitative and quantitative)Column: 30 m × 0.25 mm I.D., × 0.25 mmMedium: Helium as carrier gas, flow rate of 1 mL/minIsopinocarveol, α-ylangene, 1H-3a,7-Methanoazulene, octahydro-tetramethylCaryophyllene, trans-Z-α-Bisabolene epoxide, Benzene, 1-(1,5-dimethyl-4-hexenyl)-4-methyl- trans-α-Bergamotene, β-Bisabolene, β-Cubebene cubedol Sativen Methyl-hexadecatetraenoate, Alloaromadendrene oxide-(1) αacorenol, epi-cis sesquisabinene hydrate Octadecadiynoic acid, methyl ester, Phenol, 2-methyl trimethylcyclopentyl)-(S)-Isopropyl-2,8-dimethyl-9-Oxatricyclo decan-7-one, Hexadecanoic acid, ethyl ester, Octadecynoic acid[97]48*Trigonella foenum-graecum* (Hairy roots)GC-MS (Qualitative and quantitative)Column: (20 m × 0.20 mm i.d., 0.5 μm film thicknessMedium: Helium as carrier gas, flow rate of 0.6 mL/min.Hexanal, γ-butyrolactone, cyclopenta-1,2-dieno, valeric acid, 3- hydroxy-4,5-dimethyl-2(5H)- furanone (sotolone), 2-(4-methylthiazo-5-yl)-ethanol, 6-methyl-2,3-dihydroxy-5,6-dihydropyran-4-one, 3-amino-4,5-dimethyl-2(5H)-furanone, 5-(hydroxymethyl) furan-2-carboxaldehyde
*Trigonella foenum-graecum* (Seed)HP_5_MS GC-MS (Qualitative)Column: 30 mmin length, 0.25 mm i.d. and 0.25 µmMedium: Helium as carrier gas, flow rate of 1 mL/min3,5-Octadiene, p-Xylene, δ-3-Carene, Limonene, Decanal, β-Thujone, Hexadecanoic acid, Cis-calamenene[98]49*Vaccinium angustifolium* (Fruit)P&T-GC-MS (Qualitative)Column: (0.25 mm × 30 m × 0.25 m)Medium: Helium as carrier gas, flow rate of 1 mL/minEthyl caprylate, Linalool, 2-Nonanone, Ethyl acetate, 2-Methylbutyraldehyde[99]50*Withania somnifera* (Root)GC-MS (Qualitative)Column: (30 m × 0.32 mm × 0.25 μm)Medium: Helium as carrier gas, flow rate of 1 mL/minOleic acid, phytol, n-hexadecanoic acid, 9-octadecenoicacid(z)-, methyl ester, hexadecanoic acid, methy ester, 2-methoxy-4-vinylphenol, azetidin-2-one3,3-dimethyl-4-(1-aminoethyl), 17-octadecynoicacid, o-bromoatropine, and sucrose[100]51*Xenostegia**tridentata*LC-ESI-MS/MS (A) water with 0.1% formic acid and solvent. (B) was acetonitrile with 0.1% formic acid.3,5-dicaffeoylquinic acid, luteolin-7-O-glucoside, quercetin-3-O-rhamnoside, kaempferol-3-O-rhamnoside.[101]52*Jasminum grandiflorum* L.HPLC-PDA-MS/MS-ESI.Column: Gemini C18 110 Å (150 × 2 mm, 5 μm)Medium: A gradient of water and acetonitrile (ACN) (0.1% formic acid each) was applied from 5% to 30% ACN Quercetin, kaempferol, myricitrin, laricitrin 3-o-glucoside, myricetin 3-xyloside, reynoutrin, kaempferitrin, oleuropein, multifloroside, isoquercitrin, oleuropein glucoside, verbascoside[102]53*Yucca gigantea*(Leaves)
LC-MS/MSColumn: X select HSS T3 (2.5 µm, 2.1 mm × 150 mm)Medium: Buffer A (5 mM ammonium formate buffer pH 3 containing 1% methanol), Buffer B (5 mM ammonium formate buffer pH 8 containing 1% methanol), and buffer C (100% acetonitrile). Composition: 90 (A or B): 10 (C) Okanin-4′-o-glucoside (marein), citraconic acid, muconic acid, spirostan-3-ol-glucoside-galactoside, hecogenin, kaempferol-7-o-neohesperidoside, kaempferol-3-o-α-l-rhamnoside, spirostan-3-ol-diglucoside, luteolin-7-o-β-d-glucoside, vitexin-2″-o-rhamnoside, hesperetin-7-o-neohesperidoside, isorhamnetin, quercetin-4′-o-glucoside, isorhamnetin-3-o-rutinoside, kaempferol-3-o-(6-p-coumaroyl)-glucopyranoside, poncirin, 4′,5,7-trihydroxyflavonol, hesperetin, naringenin, apigenin-7-o-β-d-glucoside, luteolin, caffeic acid, acacetin, 3,3′,4′,5-tetrahydroxy-7-methoxyflavone[103]54 *Cycas thouarsii* (Leaves)LC-MS/MSColumn: X select HSS T3 (2.5 µm, 2.1 mm × 150 mm)Medium: Buffer A (5 mM ammonium formate buffer pH 3 containing 1% methanol), Buffer B (5 mM ammonium formate buffer pH 8 containing 1% methanol), and buffer C (100% acetonitrile).Composition: 90 (A or B): 10 (C)Quinic acid, chlorogenic acid, trigonelline, piperidine, pantothenate, cinnamaldehyde, resveratrol, ferulic acid, quercetin 3-o-glucuronide, vitexin-2′′-o-rhamnoside, vitexin, hyperoside, luteolin-7-o-glucoside.

plants-11-03243-t002_Table 2Table 2Online resources for network biology, network pharmacology, and gene enrichment analysis.Sr. No.ResourcesBrief SummaryApplicationWebsite/URL(Access Date: 7 October 2022)1BioCartaOnline maps of metabolic and signaling pathwaysDatabase of gene interaction modelshttps://maayanlab.cloud/Harmonizome/dataset/Biocarta+Pathways2BioGRIDBiological General Repository for Interaction DatasetsRetrieval of protein–protein interaction networkhttp://thebiogrid.org/3C2MapsComputational Connectivity MapsAnnotation of drug–protein pairshttp://bio.informatics.iupui.edu/4ChEMBLDatabase of bioactive compoundsRetrieval of functional as well as binding information of active compoundshttps://www.ebi.ac.uk/chembl/5ChemProtChemical–protein–disease annotation databaseAnalysis of interaction between chemical and proteinhttp://www.cbs.dtu.dk/services/ChemProt-2.0/6ChemSpiderDatabase of chemical structuresRetrieval of chemical structureshttp://www.chemspider.com/7COGsClusters of Orthologous GeneClassification of proteins on phylogenetic basishttps://www.ncbi.nlm.nih.gov/COG/8CPDBConsensus Path DataBaseMolecular functional interaction databasehttp://cpdb.molgen.mpg.de/9CytoscapeDatabase for network construction and visualizationNetwork analysishttps://cytoscape.org/10DAVIDDatabase for Annotation, Visualization & Integrated DiscoveryFunctional annotationhttps://david.ncifcrf.gov/11DIPDatabase of Interacting proteinsAnalysis of protein–protein interaction networkhttp://dip.doe-mbi.ucla.edu12DisGeNET Database of Interacting proteinsPathway analysis(https://www.disgenet.org/)13GeneCardsDatabase of human genesFor identification of disease-related geneshttps://www.genecards.org/14GuessComputer program for the analysis and visualization of networksNetwork analysishttp://www.levmuchnik.net/Content/Networks/ComplexNetworksPackage.html15HAPPIHuman Annotated & Predicted ProteinRetrieval of protein–protein interaction networkhttp://bio.informatics.iupui.edu/HAPPI/16HPRDHuman Protein Reference DatabaseRetrieval of protein–protein interaction networkhttp://www.hprd.org/17InterProIntegrative database of protein familiesCollection of protein familieshttp://www.ebi.ac.uk/interpro/18KEGGKyoto Encyclopedia of Genes and GenomesPathway analysishttp://www.genome.jp/kegg/19MetaCoreTMMetaCore (TM)Pathway analysishttp://www.genego.com20Metascape Computer program for the analysis and visualization of networksNetwork and Pathway analysis(https://metascape.org)21NetMinerComputer program for the analysis and visualization of networksNetwork analysishttp://graphexploration.cond.org/22NetPathNetwork pathway analysisPathway analysishttp://www.netpath.org/23NetworkXComputer program for the analysis and visualization of networksNetwork analysishttp://www.analytictech.com/ucinet/24Network analyst Computer program for the analysis and visualization of networksNetwork analysis(https://www.networkanalyst.ca/), 25OPHIDOnline predicted human interaction databaseRetrieval of protein–protein interaction networkhttp://ophid.utoronto.ca26PajekComputer program for the analysis and visualization of networkNetwork analysishttp://pajek.imfm.si/doku.php27PDBProtein Data bankRetrieval of protein-related informationhttp://www.rcsb.org/pdb/28PharmGBKPharmacogenomics knowledge baseAnalyze the genes’ response to drugshttp://www.pharmgkb.org/29ReactomeDatabase of pathways, reactions, and biological processesPathway analysishttp://www.reactome.org30SignaLinkSignaling pathway analysis resourcePathway analysishttp://signalink.org/31STITCHSearch Tool for Interactions of ChemicalsAnalysis of target–drug relationship and biological pathwayshttp://stitch.embl.de/32STRINGSearch Tool for the Retrieval of Interacting Genes/ProteinsRetrieval of protein–protein interaction networkhttp://string-db.org/33SwissTargetPredictionEstimate the macromolecular targets of a small moleculeIdentification of compound-related geneshttp://www.swisstargetprediction.ch/34UcinetComputer program for the analysis and visualization of networksNetwork analysishttp://www.netminer.com/35UniProtKBUniversal protein knowledge databaseAnalysis of proteinhttp://www.uniprot.org/uniprot/
plants-11-03243-t003_Table 3Table 3Network pharmacology-based investigation of multi-targeted effects or molecular mechanisms of some herbal medicines or their derived products.Sr. No.Medicinal Plant/Traditional FormulationsPhytoconstituentsBiomolecular Targets/GenesTherapeutic RoleMechanism of ActionReferences1*Boerhaavia diffusa*Apigenin, ferulic acid and quercetin ABCA1, AKT1, MMP9, LEPR, AKR1C3, KDR, PPARs, ESR1, ARNT, PON2, SESN2, IRS1, ILs, MAPKs, CAPSs, NOS, etc.Nephroprotective Reduction of oxidative and inflammatory stress, glomerulonephritis, reduction of vascular rigidity in hypertension[33]2*Tinospora cordifolia*Gallic acid and rutinCASPs, MAPKs, NOS, PRNP, CTGF, SREBF1, and JUNNephroprotectiveReduction of oxidative and inflammatory stress, glomerulonephritis[33]3*Momordica charantia*Linalool, quercetin, gallic acid, apiole, ferulic acid, caffeic acid, limonene and catechinNOS, ILs, CASPs, MAPKs, G6PD, MAPK1, MAPK3, MMPs, PTGS2, PON1, ILs, AKT1, JUN, MMPs, PPARG, TP53, etc.Diabetic nephropathyAmelioration in endothelial dysfunction, fatty liver disease, diabetes mellitus, acute kidney injury, fibrosis, hypertensive disease, obesity, etc.[34]4-Quercetin and kaempferolTNF, JUN, IL6, STAT3, MAPK1, and MAPK3Acne VulgarisAnti-inflammatory effect and regulate the excessive lipogenesis in sebaceous glands via different signaling pathways[104]5*Gelsemium elegans*Gelegamine E, gelsesyringalidine, humantenine, gelsedine, 19-z-akuammidine, gelegamine B, tabersonine, koumine, gelsemineMAPK3, HSP90AA1, JUN, EGFR, CDK1, TNF, CCND1, ESR1, PRKACA, CCNA2, CDC25C, CDK2, CCNB1, AR, CREBBP, AURKA, CDC25A, CHEK1, BCL2L1, and PIK3CDAnti-cancer Pathways in cancer, cell cycle, and colorectal cancer[105]6Danning Tablets (A polyherbal formulation)Quercetin, ß-sitosterol, luteolin, kaempferol, supraene, curcumenolactone C, and stigmasterolIL6, MAPK8, VEGFA, CASP3, ALB, APP, MYC, PPARG, and RELANonalcoholic Fatty Liver DiseaseLipid metabolism, inflammation, oxidation, insulin resistance (IR), atherosclerosis, and apoptosis[106]7*Panax notoginseng*Mandenol, beta-sitosterol, stigmasterol, ginsenoside rh2, ginsenoside f2, quercetinVEGFA, MMP-9, MMP-2, FGF2, and COX-2Diabetic RetinopathyIntervention of angiogenesis, inflammation, and apoptosis[107]8*Angelica sinensis*Isorhamnetin, jaranol, 3,9-di-O-methylnissolin, mairin, formononetin, isoflavanone, quercetin, kaempferol and (3R)-3-(2-hydroxy-3,4-dimethoxyphenyl)chroman-7-olVEGFA, TP53, IL-6, TNF, MARK1Diabetic nephropathyApoptosis, oxidative stress, inflammation, glucose, and lipid metabolism processes[108]9*Caesalpinia pulcherima*Gallic acid, catechin, ellagic acid, quercetina and cyanidin 3-glucosideESR-1, ESR-2, ESRRA, MET, VEGF, FGF, PI3K, PDK-1, MAPK, PLK-1, NEK-2, and GRKBreast cancerBreast cancer, endometrial cancer, and osteoporosis, estrogen signaling pathway, prolactin signaling pathway, thyroid hormone signaling pathway, and relaxin signaling pathway[109]10Chinese herbal formula Shenyi (SY)Moupinamide, quercetin, eriodictyol, hesperetin, formononetin, palmatine, norlobeline, taxifoline, sinoacutine, beta-sitosterol, kaempferol, stigmasterol, acacetin, linarin, and isovitexinAKT1, TNF, IL6, TP53, VEGFA, EGFR, CASP3, JUN, IL1B, MYC, ESR1, HIF1A, HSP90AA1, EGF, PTGS2, MMP9, and CCND1Diabetic nephropathyGlomerulosclerosis, nephrotoxicity, oxidative stress, inflammation, and hyperglycemia, inhibit RhoA/Rho kinase to ameliorate kidney injury and decrease fibrosis[110]11*Jasminum grandiflorum* L.Quercetin, kaempferol, myricitrin, laricitrin 3-o-glucoside, myricetin 3-xyloside, reynoutrin, kaempferitrin, oleuropein, multifloroside, isoquercitrin, oleuropein glucoside, verbascosideMKK4, MKK7, I-CAM 1, IL-6, TNF, TRAF2, PI3K-Akt, MAPK, EGFR, NephroprotectiveReduction in tubular cell lining vacuolation and intratubular cast deposition in the renal tubules, oxidative stress, inflammation, and apoptosis[111]

## 4. Future Perspectives

Quality, efficacy, and safety issues about herbal medicines often concern researchers or the healthcare system due to their complex phytochemical matrix responsible for exhibiting the pharmacological effect. In a plant matrix, it is challenging to justify which type of phytochemicals is responsible for exhibiting the pharmacological effect and to what extent. Besides that, herbal medicines and their derived products still represent the need for their comprehensive safety concern. Studying the phytochemicals of medicinal plants using modern analytical techniques such as LC-MS, GC-MS, and NMR, etc., are the most precise, accurate, and robust techniques for the qualitative and quantitative-based standardization of medicinal plants. These techniques have been exponentially used to explore the metabolomics pattern of medicinal plants. Furthermore, bioassay-based exploration of metabolomics patterns of medicinal plants brings to reach several facts concerning efficacy and safety. Qualitative and quantitative phytochemicals-based exploration of medicinal plants leads to unfolding their authentication concerns and provides accuracy in the regulatory-based exploration of medicinal plants. In recent times, several types of research have been conducted to explore the metabolomic pattern of medicinal plants using single technology such as LC-MS, GC-MS, NMR, etc. As we are aware of the complexity of the phytochemicals and their different chemical nature, single analytical methods cannot be accurate and precise enough to explore the complete phytochemical profile of the compounds. Plant metabolomics is well explored using different analytical methods based on the targeted phytochemical’s nature.

With the expansion of research and development in medical technologies, more animals are being employed in the study. All throughout the world, millions of experimental animals are utilized each year. Animals’ suffering, mortality, and misery during scientific studies have long been a topic of controversy. Animal research has several critical and concerning drawbacks outside the main ethical issue, such as the need for trained labor, lengthy protocols, and high expense. In order to address the problems with animal testing and prevent unethical practices, several alternatives have been presented. For the use of animals in laboratories, the three Rs (3Rs: reduction means minimum use of animals, refinement means minimum pain or distress to the animals, and replacement means employing the use of non-sentient material) are being used as a strategy [112].

The different fundamental principles of biology can be better understood by computational methods. In order to create novel drugs, specialized computer models and software are used. Without dissecting animals, computer-generated simulations are used to estimate the numerous potential biological and toxic consequences of a chemical or prospective medication candidate. For in vivo testing, only the most promising compounds discovered via initial screening are employed. For instance, in vivo testing is required to determine a drug’s receptor binding location. The receptor binding site for a prospective therapeutic molecule is predicted using computer software called Computer Aided Drug Design (CADD) [112]. By identifying potential binding sites, CADD prevents the testing of undesired compounds with little biological activity. Also, with the help of such software programs, we can tailor-make a new drug for the specific binding site, and then in the final stage, animal testing is done to obtain confirmatory results [113,114,115,116].

In considering the 3Rs strategy, computational or in-silico approaches remain an exponential contribution to drug discovery and development, as well as drug repurposing via approaching biomolecular targets for potential therapeutic effects. Network pharmacology is one of the most efficient and conventional approaches to exploring the multi-targeted and therapeutic effects of chemicals. It is well-defined in the exploration of polypharmacology of medicinal plants or their derived products, even exponentially used to validate their traditional therapeutic claim via approaching different biological targets. However, the traditional drug research approaches are not suitable enough to study the biomolecular therapeutic application of complex components with many targets. Network biology or network polypharmacology provides a novel research technique for studying several active ingredients with their and predicting effector targets and effector mechanisms involved in the treatment of certain diseases.

Different methods and alternatives to organisms provide a contributing hand concerning the computational approach, especially network biology, and polypharmacology is well enough for drug discovery and development via approaching their biomolecular and multi-targeted effects. Moreover, in the establishment or validation of the therapeutic effect of traditional medicinal plants or any herbal formulae, metabolomics and network pharmacology provides a contribution to validate them based on multi-targeted and therapeutic effect. The findings provided by the network pharmacology for predicting effector targets and effector mechanisms are based on reported databases. It would be an effective and alternative tool for the biological or therapeutic exploration of active constituents, thus contributing to scientific validation of medicinal plants concerning their safety and efficacy.

## 5. Conclusions

This review enlightens the systematic approach of metabolomics and network pharmacology for multi-mechanistic and therapeutic exploration of traditional Indian medicinal plants or active constituents, as well as to identify biomolecular targets concerning pathophysiological alterations. In Network biology, gene enrichment analysis is generally used to predict the pathways involved in diseases. Through this study, the synergism between numerous components, targets, and pathways may also be predicted, validating the mechanistic evidence to support clinical practice and pharmaceutical impact. It also provides ideas for the establishment of further scientific evidence. Although this approach provides a theoretical basis for better clinical development of drugs for the treatment of any acute and chronic ailments, the research conducted on network pharmacology is based on database mining research, and the mechanism of drug action is only predicted. Considering the facts of metabolomics and network pharmacology, it can be one of the most promising computational techniques for the exploration of multi-therapeutic effects in both the assessment of drug development, pre-clinical or post-clinical where a drug or drug formulae are having numbers of active constituents.

## Figures and Tables

**Figure 1 plants-11-03243-f001:**
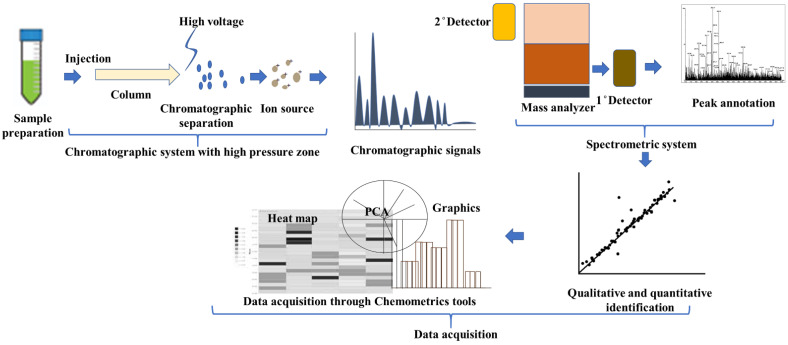
A systematic approach for metabolomic analysis and data acquisition analysis through statistical tools for medicinal plants and their derived products.

**Figure 2 plants-11-03243-f002:**
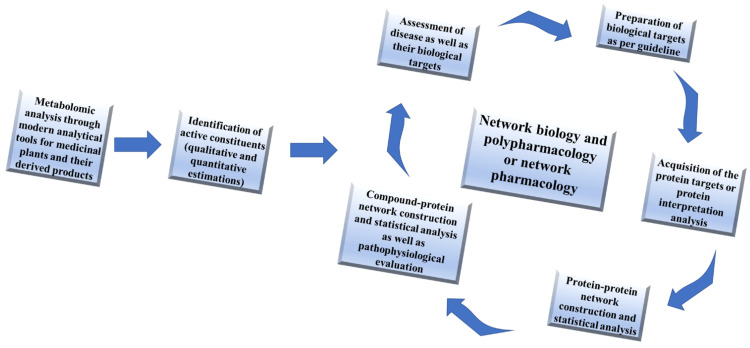
A systematic approach for network biology as well as polypharmacology.

**Figure 3 plants-11-03243-f003:**
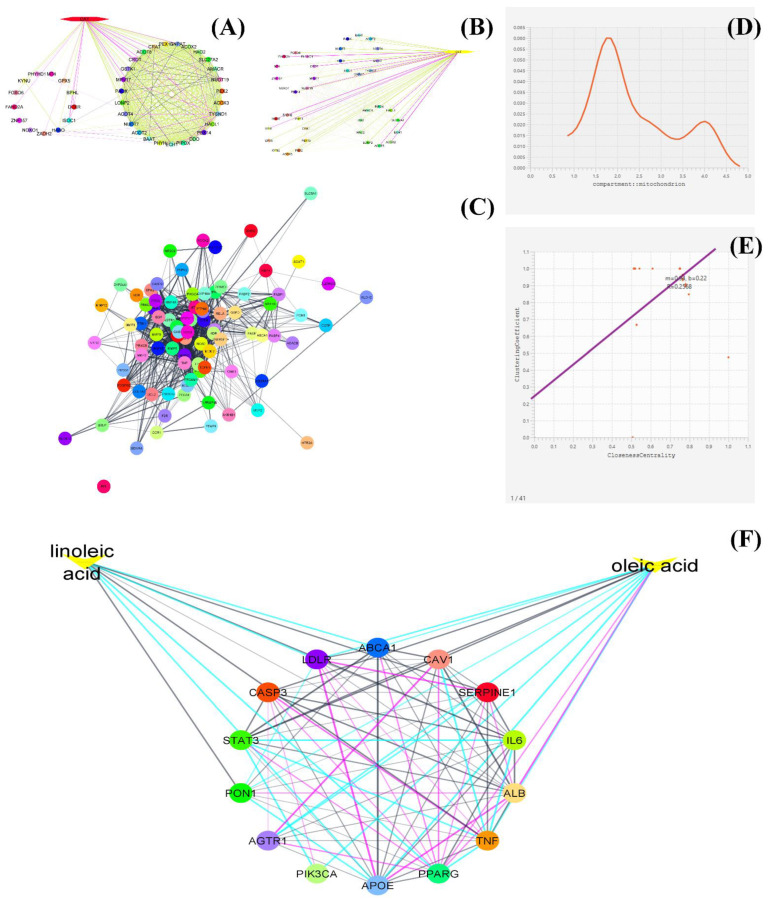
A systematic approach for network biology and pharmacology of phytochemicals, (**A**) represents the untargeted genes PPI network with partial interaction (small circle) and significant interaction (big circle), (**B**) represents selective PPI network with the specific protein (CAT) while (**C**) represents the common PPI network with partial and significant interaction of proteins/genes. (**D**) represents the histogram of the analyzed proteins based on their enrichment concerning cellular compartments (mitochondria), while (**E**) represents the regression analysis of the analyzed genes between clustering coefficient and closeness centrality. (**F**) represents the CPI interaction network which represents the interaction of two phytoconstituents with specific genes in the form of different colored edges. Each color of the edges represents a different database source. Each figure was generated as an example through the Cytoscape tool.

**Figure 4 plants-11-03243-f004:**
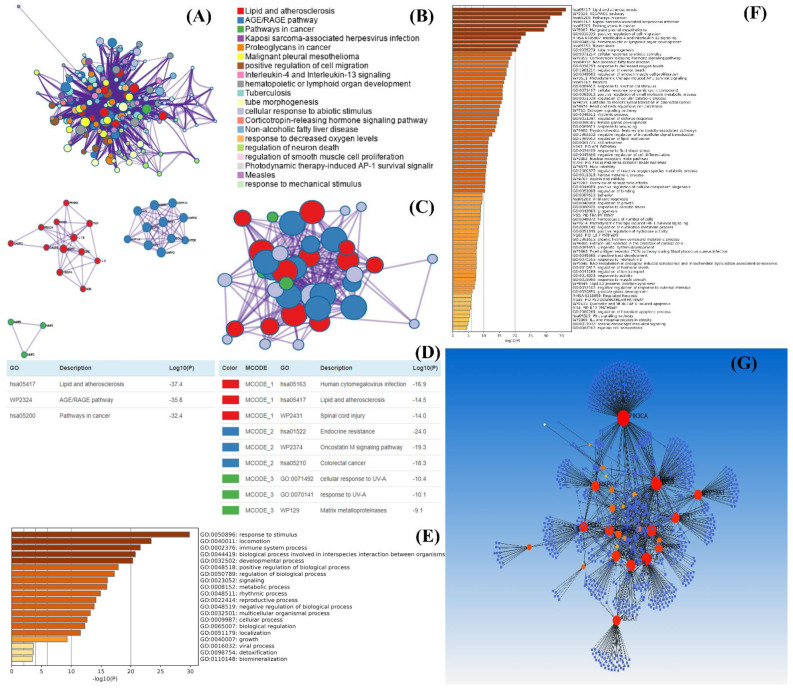
Representation of gene and disease association analysis as part of gene ontology analysis. (**A**,**B**) represent the enrichment of the analyzed genes and their pathophysiological role through different pathways, (**C**) represents the enrichment of genes based on their pathophysiological role, while (**D**) represents the enrichment scale of the genes and their role in diseases. (**E**) represents the gene ontology analysis of the selective genes, and (**F**) represents the DisGeNET analysis. This analysis was conducted as an example through the Metascape tool. (**F**) represents the gene and disease association network among different genes while (**G**) represents large nodes with high enrichment of the gene with nodes of diseases. This analysis was conducted through the Network analyst tool.

## Data Availability

Not applicable.

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
