# Peer review of "Metabolomics and Network Pharmacology in the Exploration of the Multi-Targeted Therapeutic Approach of Traditional Medicinal Plants"

_plants, 2022, doi:10.3390/plants11233243_

Round 1

Reviewer 1 Report

The review entitled ''Metabolomics and network pharmacology in the exploration of the multi-targeted therapeutic approach of traditional medicinal plants'' is interesting and organized. However, the authors need to make some changes.

·       Before the title, please write ''review'' instead of ''article'' to clarify the type of your manuscript.

·       Use more keywords such as '' phytochemical profiling, LC-MS '' to increase the visibility and readability of your article.

·       The authors should cite more recent the reference, especially in the introduction to cover the relevant literature, such as

-Metabolic Profiling of Jasminum grandiflorum L. Flowers and Protective Role against Cisplatin-Induced Nephrotoxicity: Network Pharmacology and In Vivo Validation, Metabolites 12(9) (2022)

- Elucidation of the Metabolite Profile of Yucca gigantea and Assessment of its Cytotoxic, Antimicrobial, and Anti-Inflammatory Activities, Molecules 27(4) (2022)

-Antibacterial activity and wound healing potential of Cycas thouarsii R. Br n-butanol fraction in diabetic rats supported with phytochemical profiling, Biomedicine & Pharmacotherapy 155 (2022)

·       In the last paragraph of this section Page 4 ''3.1. Need of metabolomic study for medicinal plants'', the authors wrote the name of scientists without reference.

·     In the section ''3.1.1. Challenges in metabolomics analysis'', please combine the scattered sentences to make relative paragraphs. Each paragraph should contain at least 4 or 5 sentences.

·       Please, use a white background instead of the dark one in Figure 1. A and clarify if this figure has any copyrights or if you design it.

·       Please, adjust the table data, some bold and some not, some aligned center and some left ….. also adjust the column width to cope with the data inside.

·       The resolution of the Figures is unsatisfactory. Could you please increase the resolution and use a larger size?

·       Figure 4. Letters used as (D) and (E) above the data of the figure; please adjust.

·        Please, check the abbreviations throughout the manuscript.    

·    In general, the representation of Figures and Tables needs improvements.

·       The review conclusion should contain criticized comments. Also should contain clear, specific future expectations and recommendations.

Author Response

COMMENTS AND THEIR RESPONSE

Reviewer 1

The review manuscript entitled ''Metabolomics and network pharmacology in the exploration of the multi-targeted therapeutic approach of traditional medicinal plants'' 

  •  

S. No

Comments

Response

1.       

Before the title, please write ''review'' instead of ''article'' to clarify the type of your manuscript.

The title of the manuscript has been changed as per suggestion.

2.       

Use more keywords such as '' phytochemical profiling, LC-MS '' to increase the visibility and readability of your article.

Keywords have been increased accordingly to increase the visibility and readability of your article.

3.       

The authors should cite more recent the reference, especially in the introduction to cover the relevant literature, such as

-Metabolic Profiling of Jasminum grandiflorum L. Flowers and Protective Role against Cisplatin-Induced Nephrotoxicity: Network Pharmacology and In Vivo Validation, Metabolites 12(9) (2022)

- Elucidation of the Metabolite Profile of Yucca gigantea and Assessment of its Cytotoxic, Antimicrobial, and Anti-Inflammatory Activities, Molecules 27(4) (2022)

-Antibacterial activity and wound healing potential of Cycas thouarsii R. Br n-butanol fraction in diabetic rats supported with phytochemical profiling, Biomedicine & Pharmacotherapy 155 (2022)

The relevant articles have been cited throughout the manuscript as per suggestion.

4.       

In the last paragraph of this section Page 4 ''3.1. Need of metabolomic study for medicinal plants'', the authors wrote the name of scientists without reference.

The typo error has been corrected, accordingly.

5.       

In the section ''3.1.1. Challenges in metabolomics analysis'', please combine the scattered sentences to make relative paragraphs. Each paragraph should contain at least 4 or 5 sentences.

The necessary changes have been conducted as per suggestion.

6.       

Please, use a white background instead of the dark one in Figure 1. A and clarify if this figure has any copyrights or if you design it.

Figure 1 has been corrected as per suggestions.

7.       

Please, adjust the table data, some bold and some not, some aligned center and some left ….. also adjust the column width to cope with the data inside.

Table data has been adjusted properly as per suggestion.

8.       

The resolution of the Figures is unsatisfactory. Could you please increase the resolution and use a larger size?

Resolution of the figures has been increased as much as possible.

9.       

Figure 4. Letters used as (D) and (E) above the data of the figure; please adjust.

Figure 4. Letters used as (D) and (E) above the data of the figure has been adjust.

10.   

Please, check the abbreviations throughout the manuscript.    

Abbreviations throughout the manuscript have been checked.

11.   

In general, the representation of Figures and Tables needs improvements.

Figures and Tables have been improvements.

12.   

The review conclusion should contain criticized comments. also should contain clear, specific future expectations and recommendations.

Conclusion section has been improved with clear comments and specific future expectations and recommendations.

Thanks and regards

Dr. Dinesh Kumar Yadav

Professor

Department of Pharmacognosy

SGT College of Pharmacy,

SGT University, Gurugram Haryana, India-122505

Mob: 7042348251

Reviewer 2 Report

Remarks: In Table 1, information about Solanum nigrum appears in 2 positions (4 and 13). Wouldn't it be better if they were cumulative? Likewise Trigonella foenum-graecum (position 6 and 14).

Also in table 1, in column 2, should be the name of the species and in brackets the useful part [eg. Artemisia absinthium (Aerial

part)], not so "Glycyrrhizae radix (root)" (heading 21). "Glycyrrhizae radix" is not the name of the species, but the useful part. Ditto position 21. In positions 27 and 56 the information is not complete. At position 54 there is no species.

In scientific names, the second word starts with a small letter, example (page 13, 16, 17) "Momordica Charantia", correct Momordica charantia.

Author Response

COMMENTS AND THEIR RESPONSE

Reviewer 2

The review manuscript entitled ''Metabolomics and network pharmacology in the exploration of the multi-targeted therapeutic approach of traditional medicinal plants'' 

S. No

Comments

Response

1.       

In Table 1, information about Solanum nigrum appears in 2 positions (4 and 13). Wouldn't it be better if they were cumulative? Likewise Trigonella foenum-graecum (position 6 and 14).

The information about both species has been made cumulative as per suggestion.

2.       

Also in table 1, in column 2, should be the name of the species and in brackets the useful part [eg. Artemisia absinthium (Aerial part)], not so "Glycyrrhizae radix (root)" (heading 21). "Glycyrrhizae radix" is not the name of the species, but the useful part. Ditto position 21. In positions 27 and 56 the information is not complete. At position 54 there is no species.

The necessary changes have been made in the revised manuscript as per suggestion.

3.       

In scientific names, the second word starts with a small letter, example (page 13, 16, 17) "Momordica Charantia", correct Momordica charantia.

Thanks and regards

Dr. Dinesh Kumar Yadav

Professor

Department of Pharmacognosy

SGT College of Pharmacy,

SGT University, Gurugram Haryana, India-122505

Mob: 7042348251

Round 2

Reviewer 1 Report

The authors revised the manuscript according to the recommendations.